# Defining Data Model Quality Metrics for Data Vault 2.0 Model Evaluation

**Heli Helskyaho** [1,2] , **Laura Ruotsalainen** [2,*] **and Tomi Männistö** [2]

1 Miracle Finland Oy, 00580 Helsinki, Finland; heli.helskyaho@miracleoy.fi
2 Faculty of Science, Department of Computer Science, University of Helsinki, 00014 Helsinki, Finland; tomi.mannisto@helsinki.fi
* Correspondence: laura.ruotsalainen@helsinki.fi

**Abstract:** Designing a database is a crucial step in providing businesses with high-quality data for decision making. The quality of a data model is the key to the quality of its data. Evaluating the quality of a data model is a complex and time-consuming task. Having suitable metrics for evaluating the quality of a data model is an essential requirement for automating the design process of a data model. While there are metrics available for evaluating data warehouse data models to some degree, there is a distinct lack of metrics specifically designed to assess how well a data model conforms to the rules and best practices of Data Vault 2.0. The quality of a Data Vault 2.0 data model is considered suboptimal if it fails to adhere to these principles. In this paper, we introduce new metrics that can be used for evaluating the quality of a Data Vault 2.0 data model, either manually or automatically. This methodology involves defining a set of metrics based on the best practices of Data Vault 2.0, evaluating five representative data models using both metrics and manual assessments made by a human expert. Finally, a comparative analysis of both evaluations was conducted to validate the consistency of the metrics with the judgments made by a human expert.

**Keywords:** data warehouse; Data Vault 2.0; data model; metrics





## 1. Introduction

The term data warehouse (DW) was first coined in 1990 [1]. DW was then defined as the subject-oriented, integrated, time-variant, and non-volatile collection of data in support of a decision-making process in management. In 2000, a new, modern DW methodology called Data Vault was introduced. A DW database is created using data modeling techniques, such as Inmon's, Kimball's, or the Data Vault approach. These three approaches are compared in [2,3].

An essential part of the DW is the database. The database should store correct data for decision making. To guarantee this, data modeling techniques are used. In the Data Vault 2.0 methodology, the raw data vault layer consists of three main entity types: Hubs, Links, and Satellites. A Hub entity represents a business concept identified by a business key. A Link is a unique list of relationships, associations, events, or transactions between two or more business keys. Satellites include data over time [4–8].

The quality of a data model directly correlates with data quality: database designing is the key to good-quality data that enable correct and efficient decision-making for businesses. The evolution of software development, programming languages, increasing amounts of data, different data models, and different data sources have emphasized the importance of database design to furnish precise data for informed decision-making. Designing databases manually is a time-consuming task that requires special skills and knowledge. If this process can be automated, it would enable the faster creation of good-quality databases and good-quality data.

One option for automating the process of database designing is using Generative AI and Large Language Models (LLMs). LLMs can be used to generate Data Definition

Language (DDL) specifications for data model creation. Based on experiments [4], it seems that LLM-based tools, for example, ChatGPT, are able to generate usable DDLs, but they cannot generate DDLs that follow the prescribed rules and best practices of Data Vault 2.0 methodology [5–8]. Due to the inherent non-deterministic characteristics of Language Models (LLMs) and their inability to produce flawless DDL representations aligning precisely with Data Vault 2.0 requisites, a verification process is needed for the generated models. The assessment of model quality typically involves a manual inspection by a human expert, a process that may take a considerable amount of time. Additionally, manually designing the model is, in many cases, more economical than generating a model and fixing it based on evaluations. Being able to evaluate the model automatically or, at very least, semi-automatically, would make automatic data model generation a more feasible option. Moreover, an assessment conducted through formal metrics is independent from that conducted by seasoned human experts and is free from the bias of their personal preferences. This evaluation process is not only faster but also has improved overall quality. There are several attempts to create metrics for measuring data model quality, even for DWs [9], but there are no metrics available for evaluating how well Data Vault 2.0 methodology is followed in the design of data models.

A Data Vault 2.0 data model must meet the principles of Data Vault 2.0 methodology to be defined as good quality. We defined a set of metrics to evaluate the quality of a Data Vault 2.0 data model against these principles. Based on these metrics, the database designer should, for example, be able to make the decision on whether a generated data model is usable with some refinement or whether creating the model manually from scratch would be less work. The designer may also utilize the evaluation results to methodologically organize data models based on their individual quality metrics. We tested our set of metrics with five Data Vault 2.0 data models, created based on a simple but relevant data source. Leveraging the set of metrics and the evaluation results, we defined a sufficient level of quality for the data model. The examination of scenarios involving the addition of new data sources to an established Data Vault 2.0 data model, as well as the application of Retrieval-Augmented Generation (RAG) or fine-tuning techniques to enhance outcomes from Large Language Model (LLM)-based tools, was beyond the scope of the current study. These considerations are reserved for future research endeavors.

This paper is organized as follows. In Section 2, we define the materials and methods used. We investigate other research in the field and justify the need for defining a new set of metrics for Data Vault 2.0 data models. Then, we define the method used for defining and evaluating new metrics. In Section 3, we define a set of metrics for Data Vault 2.0 data model quality and describe the testing of the metrics by evaluating five Data Vault 2.0 data models. First, we evaluated the data models using the set of defined metrics, and then a human expert manually evaluated them. Finally, we compared the results of these two evaluations. In Section 4, we discuss the results. Lastly, in Section 5, we present the research conclusions and delineate potential areas for future research.

## 2. Materials and Methods

This paper aims to create a set of metrics that can be used to evaluate quality of a Data Vault 2.0 data model. To achieve the research goals, we carried out the following steps:

- Investigate the literature to study existing measures and metrics;
- Define the methods used;
- Define the set of metrics for evaluating the quality of a Data Vault 2.0 data model;
- Evaluate example designs using the defined set of metrics;
- Evaluate example designs using a review by a human expert;
- Compare the result of these two evaluations.

*2.1. Previous Research on Data Model Quality and Metrics*

Traditionally, the appraisal of data model quality has primarily relied on reviews carried out either comprehensively or partially by human experts. One reason for this is that formal, quantitative measures are difficult to use in practice [10]. As the degree of automation in the quality evaluation process increases, the efficiency of the evaluations also improves. Additionally, formal, quantitative measures reduce subjective and bias factors in the evaluation process. An evaluation carried out by a human expert always includes personal preferences and is highly dependent on the person's skills and experience.

Data quality (DQ) is defined by categories and dimensions, as shown in Table 1 [11]. There are several attempts to define the metrics for data quality [12–14].

**Table 1.** Categories and dimensions of data quality [11].

| DQ Category | DQ Dimensions |
|---|---|
| Intrinsic DQ | Accuracy, Objectivity, Believability, Reputation |
| Accessibility DQ | Accessibility, Access security |
| Contextual DQ | Relevancy, Value-Added, Timeliness, Completeness, Amount of data |
| Representational DQ | Interpretability, Ease of understanding, Concise representation, Consistent representation |

The quality factors for a data model are slightly different from those for data quality. The quality factors for a data model are correctness, completeness, simplicity, flexibility, integration, understandability, and implementability [15]. Correctness measures whether the model follows the rules of the data modeling technique. This factor includes diagramming conventions, naming rules, defining rules, and rules of composition and normalization. Completeness means that the data model contains all user requirements. Simplicity is characterized by the inclusion of only the essential entities and relationships, minimizing unnecessary complexity. Flexibility defines how easily the data model can cope with changes in business and/or regulatory factors. Integration measures how consistent the data model is with the rest of an organization's data. Understandability refers to how easily the concepts and structures in the data model can be understood. Implementability is measured based on the degree of ease of implementing the data model within the time, budget, and technology constraints of the project. The quality factors for a data model and the proposed metrics for each of them [15] can be found from Table 2.

**Table 2.** Quality factors for a data model and their metrics [15].

| Quality Factors | Metric |
|---|---|
| Correctness | 1. Number of violations to data modeling standards<br>2. Number of instances of entity redundancy<br>3. Number of instances of relationship redundancy<br>4. Number of instances of attribute redundancy |
| Completeness | 5. Number of missing requirements (Type I errors)<br>6. Number of superfluous requirements (Type II errors)<br>7. Number of inaccurately defined requirements<br>8. Number of inconsistencies with process model |
| Integrity | 9. Number of missing business rules<br>10. Number of incorrect business rules<br>11. Number of business rules inconsistent with process model<br>12. Number of business rules redundantly defined in process model rules |
| Flexibility | 13. Number of data model elements which are subject to change<br>14. Probability adjusted cost of change<br>15. Strategic impact of change |

**Table 2.** *Cont.*

| Quality Factors | Metric |
|---|---|
| Understandability | 16. User rating of understandability<br>17. User interpretation errors<br>18. Application developer rating of understandability<br>19. Subject area–entity ratio<br>20. Entity–attribute ratio |
| Simplicity | 21. Number of entities (E)<br>22. System complexity (E + R)<br>23. Total complexity (aE + bR + cA) |
| Integration | 24. Number of data conflicts with Corporate Data Model<br>25. Number of data conflicts with existing systems<br>26. Number of data items duplicated in existing systems or projects<br>27. Rating of ability to meet corporate needs |
| Implementability | 28. Development cost estimate<br>29. Technical risk rating |

These metrics are empirically evaluated [10]. The empirical evaluation found only metrics 22, 26, and 28 to be useful and usable. On the other hand, it revealed two additional metrics: metric 30, which assesses Reuse Level, and metric 31, which quantifies the Number of Issues categorized by the Quality Factor. Numerous metric assessments have consistently indicated the challenge of identifying metrics that are both effective and applicable in real-world scenarios [10,16–22].

The presently established metrics possess the potential for evaluating the data vault methodology in its capacity as a modeling technique, or as an individual data warehouse (DW) solution. However, they do not inherently measure the extent to which best practices and rules outlined in the Data Vault 2.0 methodology are followed. The methodology works if and only if these best practices are followed. The quality factor of correctness emerges as the most proximate measure for this purpose, yet it lacks the granularity required to assess compliance with the specific standards of Data Vault 2.0.

The metrics proposed to measure data warehouse data model quality in the literature was investigated using Systematic Literature Review (SLR) process [9]. It was discovered that researchers have used theoretical and empirical methods for the validation of the metrics created. For theoretical validation, they used theory framework, axiomatic approach, or DISTANCE framework. Empirical validation techniques included categories of techniques, such as non-parametric correlational analysis, parametric analysis, regression analysis, and machine learning. In the investigation, several suggested metrics for evaluating DW data model quality [23–26] were found, but these metrics are to be used with star schema modeling techniques, including fact and dimension tables, not with Data Vault 2.0 techniques with Hubs, Links, and Satellites. Hence, these metrics are not directly applicable to the Data Vault 2.0 methodology in its entirety. We employed similar ideas and techniques to adapt the metrics for compatibility to Data Vault 2.0 modeling.

### 2.2. Defining the Method Used

The process of defining metrics should be carried out in a methodological manner: starting with the metric candidate definition followed by both its theoretical and empirical validation [23,27,28]. A theoretical validation can be conducted using axiomatic approaches or measurement theory. An empirical validation step is carried out to prove the practical utility of the metric. The empirical validation can be carried out using experimentation or case studies. The process of defining and validating metrics is evolutionary and iterative. Based on the feedback from validations, both theoretical or empirical, metrics can be redefined or discarded.

To define the set of metrics for evaluating a Data Vault 2.0 data model, we will reference the Data Vault 2.0 methodology guidelines [5–8] to identify rules that are particularly adept at discerning the quality of the model. The objective is to establish a set of metrics that will help to identify a model that needs as little as possible manual work from the database designer. Alternatively, the aim is to provide the database designer with a tool for evaluating multiple models and selecting the optimal design among alternative options.

We used ChatGPT 3.5 to generate the DDLs for the Data Vault 2.0 data model based on the source database DDLs. Then, we imported the DDLs to Oracle SQL Developer Data Modeler to be able to visualize the model and manually test the set of metrics. Oracle SQL Developer Data Modeler can also be used to fix the mistakes in the generated model.

## 3. Results

### 3.1. Defining Metrics for Data Vault 2.0 Data Model Quality Evaluation

In our previous research [4], it was noticed that the models generated by ChatGPT can be poor in quality, resulting in more work for the database designer than if the model was manually created. Instead of simply designing the data model, the database designer now needs to check the generated model to find errors and then to fix them. The primary objective of this research is to identify a comprehensive set of metrics for assessing the quality of the Data Vault 2.0 data model. For instance, the intention is to assist the database designer in determining the utility of a Data Vault 2.0 model generated by ChatGPT relative to a manually created counterpart.

The issues causing most of the refactoring work in a data model are typically missing tables, primary keys (PK), or foreign keys (FK). Also, missing columns are a problem, because, even though adding them might be easy, spotting that they are missing can be time consuming. Data Vault 2.0 includes a set of technical columns that each Hub, Link and Satellite should have. If those columns are missing, or the datatype is wrong, it is not a time-consuming task to fix them, since it is a straightforward task that can be efficiently programmed using Oracle SQL Developer Data Modeler [29].

Based on this understanding and the Data Vault 2.0 methodology [5–8], we defined 20 metrics, for evaluating the data model quality of a Data Vault 2.0 model. To calculate these Data Vault 2.0 data model quality metrics, we require the measures described in Table 3.

The 20 metrics, shown in Table 4, belong to three categories: schema metrics, table/column metrics, and manually evaluated metrics. Schema metrics are used to verify that there are no missing tables, PKs, or FKs, table/column metrics are used to identify missing columns. The schema metrics are the following:

1.  CDTSHS: compares the data model generated to the source data model. All the data in the source dataset should end up either as a Hub table or a Satellite table. If the number of Hub tables is less than the number of tables in the original dataset, there should be more Satellite tables than Hub tables. Examples of these are a Dependent Child or a business concept that already has a Hub table in the Data Vault model.
2.  RoT1: the ratio of the number of Satellite tables and Hub tables. The number of Satellite tables must be equal or greater than the number of Hub tables, since every Hub must have at least one Satellite, but it can have several Satellites.
3.  RoT2: the ratio of the number of Hub tables and Link tables. The number of Hub tables must be greater than the number of Link tables.
4.  RPK: the ratio of the number of tables and PKs. Every table (Hub, Link, Satellite) must have a primary key.
5.  MaxD: the maximum depth of the model. The Data Vault 2.0 data model can be wide, but it should never be deeper than three levels; the third level citizen is the max depth.

To verify that there are no columns missing, we used table and column metrics as follows:

6.  RPKH: the ratio of the number of PK columns on Hub tables. Each Hub table should have exactly one PK.

7. RPKL: the ratio of the number of PK columns on Link tables. Each Link table should have exactly one PK.
8. RPKS: the ratio of the number of PK columns on Satellite tables. Each Satellite table should have at least two columns for the PK, since Satellites hold the history data of a Hub or a Link. Most of the time the value is two but for multi-active satellites it is three; therefore, we define the value as being two or greater.
9. NoFKH: the number of FKs in Hub tables. Hub tables should not have any FKs.
10. RFKS: the ratio of FKs in Satellite tables. Each Satellite table should have exactly one FK.
11. RFKL: the ratio of FKs in Link tables. Each Link table should have at least two FKs.
12. RAH: the ratio of columns in Hub tables. Typically, the number of columns in a Hub table is four (PK + business key, Loaddate, Recordsource), but if the business key includes more than one column, then the number of columns in a Hub table is more than four.
13. RAL: the ratio of columns in Link tables. A Link table should include three columns (PK, Recordsource, Loaddate) and the FK columns (minimum two).
14. RAS: the ratio of columns in Satellite tables. A Satellite table should include two columns for the PK (the parent Hub PK + Loaddate), Recordsource, the Hashdiff column (optional), and all the actual data columns from the source dataset. The Satellite table would be useless if it did not include at least one actual data column.
15. RMPKA: the ratio of mandatory PK columns. PK columns should be defined as mandatory in all tables.
16. RMFKA: the ratio of mandatory FK columns. FK columns should be defined as mandatory in all tables.
17. RMAH: the ratio of mandatory columns in Hubs. Hub tables should only have mandatory columns.
18. RMAL: the ratio of mandatory columns in Link tables. Link tables should only have mandatory columns.

Satellite tables have optional columns if those columns are optional in the source database. Therefore, we cannot check the ratio of mandatory columns in Satellites, unless we compare them to those in the source tables. This can be carried out but the assumption is that ChatGPT follows the column definitions of the original DDLs. Based on the experiments so far, the following assumption holds.

Manually defined metrics:

19. TA, Data Vault 2.0 technical columns are correct:

    a. In Hubs (loaddate, recordsource), TAB;
    b. In Satellites (loaddate, recordsource, hashdiff), TAS;
    c. In Links (loaddate, recordsource), TAL.

20. DTTA, Data types of Data Vault 2.0 technical columns are correct:

    a. In Hubs, DTTAB;
    b. In Satellites, DTTAS;
    c. In Links, DTTAL.

The last two defined metrics are to be carried out manually. There are two reasons for this: to be able to do this in a reliable manner, we would need to follow this with naming conventions, and ChatGPT is not able to follow those without adding RAG to the process. The reply from ChatGPT is in line with this decision: "Please note that this is a simplified representation, and in a real-world scenario, you might need to consider additional aspects such as data types, constraints, indexes, and any other specific requirements of your target database platform".

We could have metrics for indexes (at least PK and FK), but since these should be a generic set of metrics and different relational database management systems (RDBMs) support and require different kind of indexing, we abstain from using indexes as an element of the set of metrics.

**Table 3.** Different measures needed to calculate Data Vault 2.0 data model quality metrics.

| Measure | Measure Description |
|---|---|
| NoTDS | Number of tables in the data source |
| NoH | Number of Hub tables |
| NoHCNoTDS | Number of Hub tables minus Number of tables in the data source (NoH-NoTDS) |
| NoS | Number of Satellite tables |
| NoL | Number of Link tables |
| NoPK | Number of PKs |
| NoFK | Number of FKs |
| NoFKH | Number of FKs in Hub tables |
| NoFKS | Number of FKs in Satellite tables |
| NoFKL | Number of FKs in Link tables |
| MaxD | Maximum number of Depth in the model |
| NoPKA | Number of PK columns |
| NoPKAM | Number of mandatory PK columns |
| NoFKA | Number of FK columns |
| NoFKAM | Number of mandatory FK columns |
| NoPKAH | Number of PK columns in Hub tables |
| NoPKAL | Number of PK columns in Link tables |
| NoPKAS | Number of PK columns in Satellite tables |
| NoFKAH | Number of FK columns in Hub tables |
| NoFKAL | Number of FK columns in Link tables |
| NoFKAS | Number of FK columns in Satellite tables |
| NoAH | Number of columns in Hub tables |
| NoAL | Number of columns in Link tables |
| NoAS | Number of columns in Satellite tables |
| NoMAH | Number of mandatory columns in Hub tables |
| NoMAL | Number of mandatory columns in Link tables |
| NoMAS | Number of mandatory columns in Satellite tables |

The criteria (equation) of these 20 metrics are shown in Table 4. Each metric from 1 to 18 is assigned a score of 1 if the specified criteria are satisfied and 0 points if they are not. Metrics 19 and 20 are established by a human reviewer and validated on a scale of 0, 0.25, 0.5, 0.75, or 1, depending on the degree to which they adhere to the Data Vault 2.0 methodology requirements. The maximum number of points for a model is 20. The metrics we have defined cover the most critical mistakes in the data model with several metrics giving them a higher weight. For example, missing tables are identified by metric 1 and, depending on the table type, by metric 2 or 3. Alternatively, missing PKs are identified using metrics 4 and 6, 7, or 8, depending on the table type.

We could define different weights for each metric to measure the amount of work needed for them, but this would make the model more complicated. This can be carried out as future research if needed, as well as using RAG to verify business keys, or to give instructions on the content of a PK column and the Hashdiff column, for example.

**Table 4.** The set of metrics created for Data Vault 2.0 data model quality evaluation and the equation/criteria of each metric.

| No. | Metric | Equation |
|---|---|---|
| 1 | CDTSHS | NoHCNoTDS = 0 or If NoHCNoTDS < 0, then NoS-Noh $\geq$ 1 |
| 2 | RoT1 | NoS/NoH $\geq$ 1 |
| 3 | RoT2 | NoH/NoL > 1 |
| 4 | RPK | (NoH + NoL + NoS)/NoPK = 1 |
| 5 | MaxD | $\leq$3 |
| 6 | RPKH | NoPKAH/NoH = 1 |

**Table 4.** *Cont.*

| No. | Metric | Equation |
|-----|--------|----------|
| 7 | RPKL | NoPKAL/NoL = 1 |
| 8 | RPKS | NoPKAS/NoS $\geq$ 2 |
| 9 | NoFKH | =0 |
| 10 | RFKS | NoFKS/NoS = 1 |
| 11 | RFKL | NoFKL/NoL $\geq$ 2 |
| 12 | RAH | NoAH/NoH $\geq$ 4 |
| 13 | RAL | NoAL/NoL $\geq$ 5 |
| 14 | RAS | If the Satellite table does not have the hashdiff column NoAS/NoS > 3, if the hashdiff column is used (recommended) then NoAS/NoS > 4. |
| 15 | RMPKA | NoPKAM/NoPKA = 1 |
| 16 | RMFKA | NoFKAM/NoFKA = 1 |
| 17 | RMAH | NoAH/NoMAH = 1 |
| 18 | RMAL | NoAL/NoMAL = 1 |
| 19 | TA | Technical columns are correct |
| 20 | DTTA | Data types of technical columns are correct |

### *3.2. Empirically Evaluating the Metrics*

The next step was to empirically test the metrics. We used the manually created example Data Vault 2.0 data model and two generated models from our previous research [4]. The generated models were from May 2023 and September 2023. A human expert reviewed the models and chose the May version with prompt engineering and the September version without it. Then, we generated a model in January 2024 with the original prompting [4] and another model using prompt engineering: additional instructions were incorporated into the prompt to address the errors identified in the initially generated DDL.

We imported the generated DDLs to Oracle SQL Developer Data Modeler one by one and investigated the results. We obtained the measures outlined in Table 3. The measures can be either visualized on the user interface of Oracle SQL Developer Data Modeler, or programmatically obtained from the data model using JavaScript. We used the visual approach. After collecting the necessary measures, we calculated the metrics, as explained in Table 4. Finally, we used the metrics to evaluate the models. We also conducted a review by a human expert to compare the evaluations by our metrics and the human reviewer.

Figure 1 illustrates the source database used [4]. The source database consists of four tables: Customers, Orders, Orderlines, and Products. This data model was chosen because it is simple, but includes the needed structures to test the main techniques used in Data Vault 2.0 modeling. It has Hubs, Links, and Satellites and it also includes a concept called Dependent Child.

Figure 2 displays an exemplar data model for Data Vault 2.0, formulated using the source data shown in Figure 1. This data model follows the best practices of Data Vault 2.0 [4]. The data model consists of nine tables: three Hub tables, two Link tables, and four Satellite tables.

We started our experiments of creating a new set of DDLs for the Data Vault 2.0 data model by prompting:

"The DDLs for the database are following:

<DDLs of the source database>

Please generate the DDLs for the target database following the Data Vault 2.0 methodology".

The data model generated by this prompt is shown in Figure 3. This data model consists of nine tables: three Hub tables, two Link tables, and four Satellite tables. The

biggest problem with the model is that the PK of Satellite tables is wrongly defined. Also, the data type of PKs is wrong.

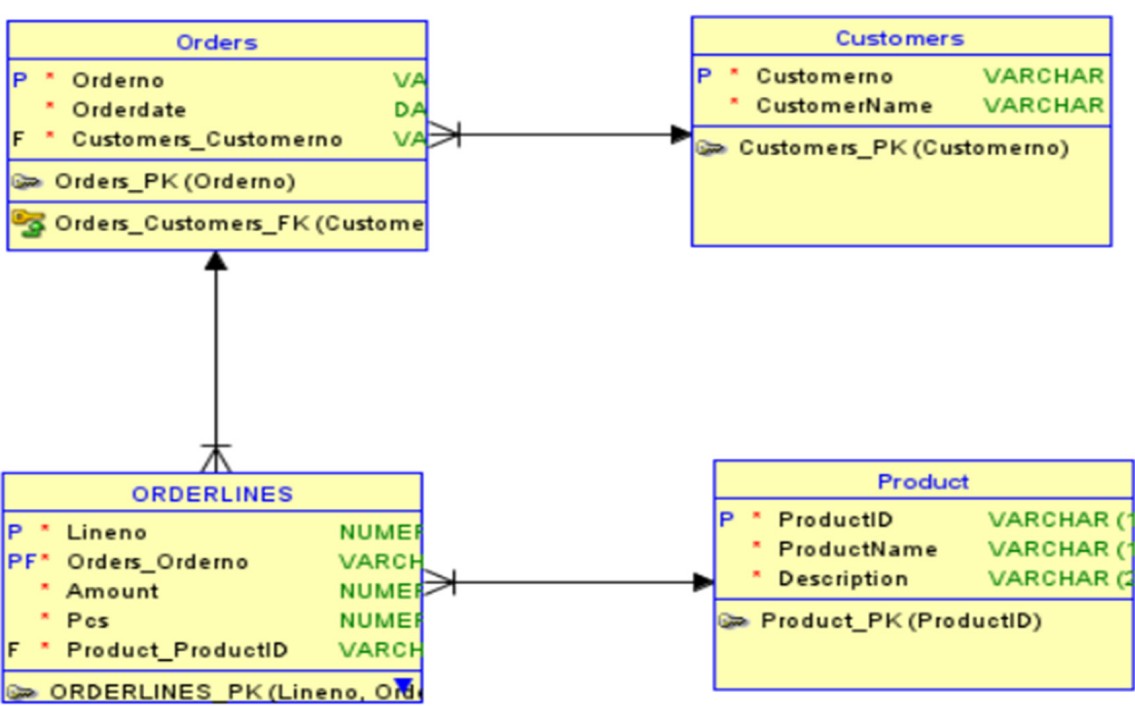

**Figure 1.** Source database for Data Vault 2.0 data warehouse creation [4].

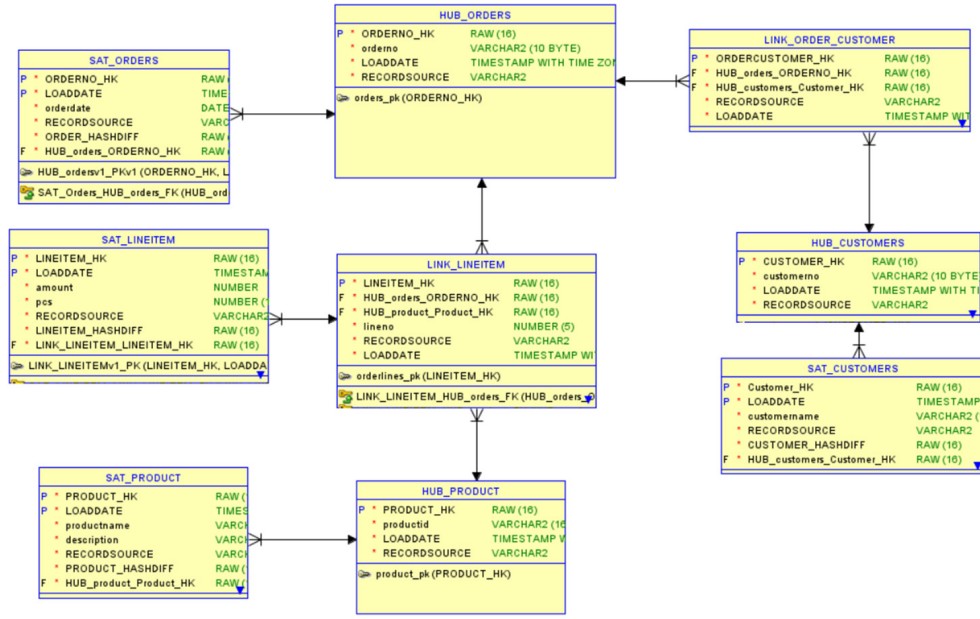

**Figure 2.** An example Data Vault 2.0 data model was manually created following the best practices [4].

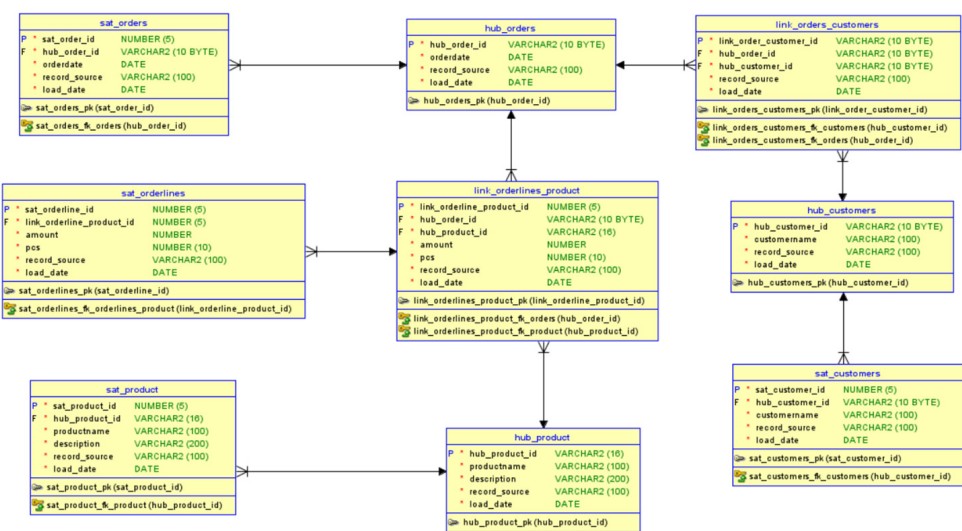

**Figure 3.** First Generative-AI-generated Data Vault 2.0 design.

Then, we prompted engineered ChatGPT to address the flaws noticed by adding in the end of the prompt:

"Also remember that all Primary keys should be hash columns of type binary. Satellites should have a hashdiff column".

The model generated from this prompt is shown in Figure 4. This data model consists of only eight tables: three Hub tables, one Link table, and four Satellite tables. Also, many FKs are missing.

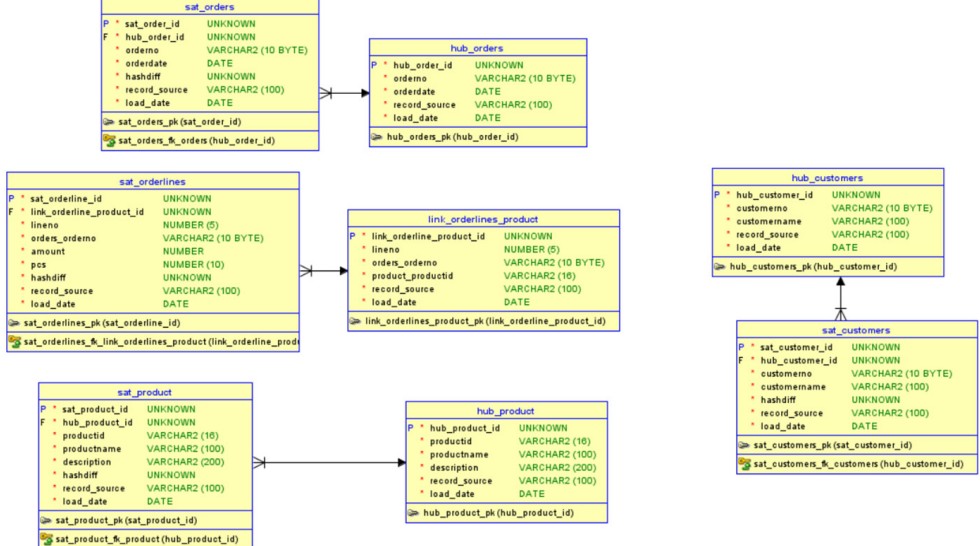

**Figure 4.** Generated Data Vault 2.0 design using prompt engineering.

For the evaluation, we used five models: the example model created manually [4], Version May 2023 with prompt engineering [4], Version September 2023 [4], Version January 2024, and Version January 2024 with prompt engineering. In the following evaluation, we refer to these models as shown in Table 5.

**Table 5.** In this table, you can see the name of a model used in the evaluations for each experiment.

| Model | Name |
|---|---|
| Example carried out manually | Model1 |
| Version May 2023 | Model2 |
| Version September 2023 | Model3 |
| Version January 2024 | Model4 |
| Version January 2024, prompt engineering | Model5 |

As shown in Table 6, we calculated all the necessary measures to obtain the metrics.

**Table 6.** The schema measures for each model.

| Measure | Model1 | Model2 | Model3 | Model4 | Model5 |
|---|---|---|---|---|---|
| NoTDS | 4 | 4 | 4 | 4 | 4 |
| NoH | 3 | 2 | 3 | 3 | 3 |
| NoHCNoTDS | −1 | −2 | −1 | −1 | −1 |
| NoS | 4 | 3 | 3 | 4 | 4 |
| NoL | 2 | 1 | 2 | 2 | 1 |
| NoPK | 9 | 3 | 8 | 9 | 8 |
| NoFK | 8 | 5 | 7 | 8 | 4 |
| MaxD | 3 | 2 | 2 | 3 | 2 |
| NoPKA | 13 | 3 | 8 | 9 | 8 |
| NoPKAM | 13 | 3 | 8 | 9 | 8 |
| NoFKA | 8 | 5 | 8 | 8 | 4 |
| NoFKAM | 8 | 0 | 8 | 8 | 4 |
| NoFKH | 0 | 0 | 0 | 0 | 0 |
| NoFKS | 4 | 3 | 4 | 4 | 4 |
| NoFKL | 4 | 2 | 4 | 4 | 0 |
| NoPKAH | 3 | 2 | 3 | 3 | 3 |
| NoPKAL | 2 | 1 | 2 | 2 | 1 |
| NoPKAS | 8 | 0 | 3 | 4 | 4 |
| NoFKAH | 0 | 0 | 0 | 0 | 0 |
| NoFKAL | 4 | 2 | 4 | 4 | 0 |
| NoFKAS | 4 | 3 | 4 | 4 | 4 |
| NoAH | 12 | 8 | 12 | 13 | 16 |
| NoAL | 11 | 7 | 12 | 12 | 6 |
| NoAS | 26 | 22 | 17 | 22 | 31 |
| NoMAH | 12 | 2 | 3 | 13 | 16 |
| NoMAL | 11 | 3 | 6 | 12 | 6 |
| NoMAS | 26 | 0 | 8 | 26 | 31 |

Then, we used these measures to calculate the metrics. The metrics for each model are shown in Table 7, where metrics with values of zero are shown in red. The red color indicates that the metric was not met.

Table 8 illustrates the models along with their respective scores, presented in both the 18-point system and 20-point system. The higher the number of points, the better the model quality.

Model5 requires much more manual work than Model4. However, the inclusion of two additional metrics in the 20-point system falsely demonstrates their equivalence. Based on this experiment, these measures should not be used, or their weights should be reconsidered. As we have acknowledged the simplicity of programmatically adding missing technical columns or correcting their data types, there is a compelling rationale to exclude metrics 19 and 20 from the set of metrics.

**Table 7.** The metrics for each model.

| Metric | Model1 | Model2 | Model3 | Model4 | Model5 |
|---|---|---|---|---|---|
| CDTSHS | 1 | 1 | 0 | 1 | 1 |
| RoT1 | 1 | 1 | 1 | 1 | 1 |
| RoT2 | 1 | 1 | 1 | 1 | 1 |
| RPK | 1 | 0 | 1 | 1 | 1 |
| MaxD | 1 | 1 | 1 | 1 | 1 |
| RPKH | 1 | 1 | 1 | 1 | 1 |
| RPKL | 1 | 1 | 1 | 1 | 1 |
| RPKS | 1 | 0 | 0 | 0 | 0 |
| NoFKH | 1 | 1 | 1 | 1 | 1 |
| RPKS | 1 | 1 | 0 | 1 | 1 |
| RFKL | 1 | 1 | 1 | 1 | 0 |
| RAH | 1 | 1 | 1 | 1 | 1 |
| RAL | 1 | 1 | 1 | 1 | 1 |
| RAS | 1 | 1 | 1 | 1 | 1 |
| RMPKA | 1 | 1 | 1 | 1 | 1 |
| RMFKA | 1 | 0 | 1 | 1 | 1 |
| RMAH | 1 | 0 | 0 | 1 | 1 |
| RMAL | 1 | 0 | 0 | 1 | 1 |
| TA | 1 | 0.5 | 0 | 0.75 | 0.75 |
| DTTA | 1 | 0 | 0 | 0 | 1 |
| **TOTAL/Score-18** | **18** | **13** | **13** | **17** | **16** |
| **TOTAL/Score-20** | **20** | **13.5** | **13** | **17.75** | **17.75** |

**Table 8.** Scores for each model using the criteria set of 18 and 20 metrics.

| Model | Score for 18 Metrics | Score for 20 Metrics [1] |
|---|---|---|
| Model1 | 18 | 20 |
| Model2 | 13 | 13.5 |
| Model3 | 13 | 13 |
| Model4 | 17 | 17.75 |
| Model5 | 16 | 17.75 |

[1] The score for metrics 19 and 20 were evaluated by a human expert on scale 0–0.25–0.5–0.75–1.

*3.3. Manual Human Expert Review of the Models*

A quick review by a human expert was performed for all models. The maximum score for a model was 20, as determined by the anticipated effort required to rectify the model. A model receiving a score of 20 points suggests that the model is suitable for use without modifications, whereas any score below 20 indicates imperfections in the model. The human reviewer uses their own expertise and experience to determine metrics.

Model1 is a data model that follows the methodology and could be used without modifications, earning a score of 20. In Model2, the Dependent Child concept was not implemented correctly; even the data attributes were lost in the model. Fixing this model would require a lot of work. The score for Model2 is 14. Model3 has similar issues as Model2, but it also has an extra FK, which causes confusion. Model3 is worse than Model2, and the score for Model3 is 13. In Model4, the PKs in the Satellite tables were incorrect. These issues can be easily addressed using Oracle SQL Developer Data Modeler; however, the errors in primary keys remain notably significant. The PKs were of the wrong data type in all tables. Fixing the PK would be easy, but that would cause changes to FKs too. With a tool, fixing would not take too long. The score for Model4 is 18. Model5 has wrongly defined PKs in Satellites, and a Link table is missing. Model5 achieves a score of 16. If we use the 18-point system, which does not include metrics for the technical columns, Model1 would achieve a score of 18, Model2 12, Model3 11, Model4 17, and Model5 15. The human reviewer would say that only Model1 and Model4 are useful models. All human expert reviews are shown in Table 9.

**Table 9.** Human reviewer scores for each model using the criteria set of 18 and 20 metrics.

| Model | Human Expert Review, Max 18 | Human Expert Review, Max 20 |
|---|---|---|
| Model1 | 18 | 20 |
| Model2 | 12 | 14 |
| Model3 | 11 | 13 |
| Model4 | 17 | 18 |
| Model5 | 15 | 16 |

### 3.4. Review Results

Both the automatic scoring using the new metrics and human expert reviews came to consistent conclusions: Model4 stands out as the best among the generated models. Based on the experiment, we conclude that metrics 19 and 20 do not bring enough value to the process, since they need to be manually checked and the flaws they indicate are easily fixable. This led to the conclusion that using the 18-point system would be better. Using the 18-point system, it seems that the minimum score defined as good quality is 17 points. Anything under that would require too much refactoring work.

### 4. Discussion

In this paper, we defined metrics for evaluating the quality of a Data Vault 2.0 data model. We defined 20 metrics and evaluated them using five example data models. A human expert also evaluated the same models. The order of the quality of the models was the same in both evaluations: the best model was the manually created example model (Model1), and the worst model was Model3. The best of the generated models was Model4. Model5 was close to the quality of Model4 but was found in the human expert review to have serious flaws that would cause extra work. We identified that two of the metrics (metrics 19 and 20) are impractical due to the requirement for manual inspection and the minor refactoring work needed to address them. Based on these evaluations, we came to the conclusion that the 18 defined metrics are useful for evaluating the quality of a Data Vault 2.0 data model and that having a minimum of 17 points is the threshold the limit for defining good quality.

### 5. Conclusions

In this paper, we defined a set of metrics for evaluating the quality of a Data Vault 2.0 data model and evaluated the metrics using a test case. We limited the scope of this paper to a simple but relevant data model. We assumed this data model to be the first data model added to a Data Vault 2.0 DW. Our assumption is that adding more data sources to an existing DW would change some of the values of the metrics. For example, the number of Hub tables might follow the Bell curve shape, since some of the required entities are already in the data model. On the other hand, these would be added as Satellite tables, increasing the number of Satellite tables. In other words, our metrics hold but their values might need some adjustments.

For future research, it would be valuable to investigate these metrics when the DW is already present and new data sources are added. Also, the model quality of a generated model could be improved using RAG or fine tuning. RAG would bring, for example, the possibility of defining naming conventions that enable the use of new metrics. The measure of collection and metric calculation could be automated. In our experimental setup, the manual method demonstrated effectiveness. Nevertheless, in practical use cases, executing the DDLs in a (test) database allows for the application of SQL queries to extract measures and obtain metric values.

**Author Contributions:** Conceptualization, H.H.; methodology, H.H.; software, H.H.; validation, H.H., L.R. and T.M.; formal analysis, H.H.; investigation, H.H.; resources, H.H.; data curation, H.H.; writing—original draft preparation, H.H.; writing—review and editing, H.H., L.R. and T.M.; visualization, H.H.; supervision, L.R. and T.M.; project administration, H.H.; funding acquisition. All authors have read and agreed to the published version of the manuscript.

**Funding:** Open access funding provided by University of Helsinki.

**Data Availability Statement:** The raw data supporting the conclusions of this article will be made available by the authors on request.

**Acknowledgments:** The authors would like to thank Dan Linstedt and Cindi Meyersohn for helpful and insightful discussions.

**Conflicts of Interest:** The company has an interest in the problem and wants to support research on solving it. All the experimental design, execution, data collection and analysis were carried out by our lab. The company agreed to the publication of the results of this study. There is no other potential conflict of interest between the funder and this study. Data Vault 2.0 as a methodology is free to use without any licenses, etc. All the results of our research are freely available. The authors declare no conflicts of interest.

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
