# Peer review of "Defining Data Model Quality Metrics for Data Vault 2.0 Model Evaluation"

_inventions, doi:10.3390/inventions9010021_

Round 1

Reviewer 1 Report

Comments and Suggestions for Authors

The document is about the introduction of new metrics for evaluating the quality of a Data Vault 2.0 data model. It addresses the lack of specific metrics for assessing adherence to Data Vault 2.0 principles and outlines a methodology for defining metrics, evaluating data models, and comparing the results with manual assessment by a human expert. The goal is to provide a more efficient and consistent way to evaluate the quality of Data Vault 2.0 data models. The methodology for evaluating data models involves defining a set of metrics for assessing the quality of a Data Vault 2.0 data model. These metrics are then used to evaluate example designs, followed by a manual assessment by a human expert. The results of these two evaluations are compared to determine the effectiveness of the defined metrics in assessing the quality of the data models. The study found that the order of quality of the models was consistent in both evaluations, with the best model being the example model created manually and the worst model being a generated one. The evaluation confirmed that the defined metrics are useful for evaluating the quality of a Data Vault 2.0 data model, and it was concluded that having a minimum of 17 points would be the limit for defining the quality as good.

The paper is well written and the paper can be accepted after considering the following points:

1.   The abstract should be modified to define the problem, its importance, the existing solutions, the limitations of these solutions and how this study avoided these limitations.

2.   Please discuss how well a data model conforms to the rules and best practices of Data Vault 2.0.

3.   How the methodology involves defining a set of metrics, evaluating representative data models using both the metrics and manual assessment aided by a human expert, and conducting a comparative analysis of both evaluations to validate the consistency of the metrics with the judgments made by a human expert?

4.   How does the evaluation process aim to make automatic data model generation a more potential option by allowing for automatic or semi-automatic evaluation of the model, thus increasing the efficiency of the evaluations and reducing subjective and bias factors in the evaluation process?

5.    What are the specific metrics introduced for evaluating the quality of a Data Vault 2.0 data model?

6.    How does the methodology for evaluating data models compare the results with manual assessment by a human expert?

7.    What are the benefits of using these new metrics to evaluate Data Vault 2.0 data models?

Reviewer 2 Report

Comments and Suggestions for Authors

Referee report on “Defining Data Model Quality Metrics for Data Vault 2.0 Model 2 Evaluation” submitted to Inventions

The article presents a methodology for building quality metrics for Data Vault 2.0 framework, considering an aggregation of several quality metrics and consistency of data models with the structure proposed by Data Vault 2.0. I believe the article is within the scope of the “Inventions and Innovation in Design, Modeling and Computing Methods" Section of Inventions, and I have some comments about the article that could be improved before possible publication.

The first comment is about the audience of the article. In its current format, the article assumes that the reader has an understanding of data models and in particular the approach more linked to computer science and data structures. In this regard, a general introduction to what the modern concept of data models and Data Vault 2.0 actually means would be useful for a wider audience, for example statisticians, economists, administrators, etc.

The second point is more conceptual – in the construction of the aggregate evaluation metric, with the results presented in Tables 8 and 9, it is implicitly assumed that each metric has the same weight in the overall aggregation. Although I understand this simple way of aggregating metrics, I believe that equal weighting is not necessarily appropriate, as different categories in the metric may represent different computational, time and economic costs for adapting the metric. So, I would like a justification or a more detailed discussion about this choice.

With these two modifications, I believe that the work has a contribution with publication potential.

Reviewer 3 Report

Comments and Suggestions for Authors

This work presents a new methodology aimed at evaluating the quality of a Data Vault 2.0 model, either manually or automatically. This methodology defines a set of metrics to apply. The paper presents an evaluation of five representative data models using both the new metrics and manual analysis performed with the help of a human expert. This work presents a clear innovation related to the quality evaluation process of a Data Vault 2.0 data model. It must be taken into account that for a design process to be carried out automatically, it is necessary to have an automatic mechanism that validates it, as well as obtaining a quantitative evaluation that allows automatically converging towards the best solution. That is why the metrics presented are an important first step to fully automate the evaluation of Data Vault 2.0 data models, and will also enable the creation of new AI-based generative methods.

General comments:

- The article is well structured and presents the theoretical bases in sufficient depth, as well as relevant references for a deeper understanding.

- All sections are clear and well developed, and the methodology is presented comprehensively. Figures and tables are descriptive.

- The paper is technically sound, and the experiments are well presented. Perhaps the variety of evidence could be greater so that the conclusions could be more precise. I think the problem with using ChatGPT in any such study is reproducibility. This AI system is constantly evolving, so for the prompts described, the answers will be different as time goes by. In any case, it is a fast and independent system to generate the tested DDLs, and of course this does not invalidate the tested scenarios.

- The article is easy to understand, although the writing could benefit from some revision as there are some minor issues.

- The bibliography is adequate, although many of the references are somewhat dated. Only 10% of the references are from the last 5 years. However, this makes some sense due to the specificity of the contribution.

- The conclusions and discussion are consistent with the results.

Specific comments:

- Although references are used, a summary of how the empirical evaluation of the metrics presented in section 2.1 was carried out would be good. This would make the paper more self-contained, because this is an important point.

- Perhaps it is a matter of custom, but I think it would be better to rephrase the titles of some sections that use the gerund.

- Personally, I would rename section 2.1 to something like "state of the art in..." or "previous work on…".

- There are some phrases that could benefit from a revision. Although most are correct, they are quite unorthodox. E.g.: “a verification process is necessitated for the generated models” -> “a verification process is needed for the generated models”.

- Likewise, some tenses would be better in the passive voice. E.g.: “…you can find the quality factors…” (in line 113).

- It seems that both British English and American English are mixed together. E.g.: “modelling” and “modeling”.

- Minor indentation issues, as in section 2.1 on line 86.

Comments on the Quality of English Language

The writing could benefit from some revision as there are some minor issues. See specific comments.

Round 2

Reviewer 1 Report

Comments and Suggestions for Authors

The authors covered all my comments, and I recommend the acceptance paper in its present case.